# Three-dimensional structural model of the Qaidam basin: Implications for crustal shortening and growth of the northeast Tibet

5 Jianming Guo<sup>1</sup>, Xuebing Wei<sup>2</sup>, Guohui Long<sup>2</sup>, Bo Wang<sup>2</sup>, and Shiyang Xu<sup>1</sup>

<sup>1</sup>Key Laboratory of Petroleum Resources, Gansu Province / Key Laboratory of Petroleum Resources Research, Institute of Geology and Geophysics, Chinese Academy of Sciences, Lanzhou 730000, China. <sup>2</sup>Research Institute of Petroleum Exploration and Development, PetroChina Qinghai Oilfield Company, Dunhuang

10 **736202, China**.

Corresponding to: Jianming Guo (gjm2001cn@yahoo.com)

Abstract. We reconstruct the main geological structures and surfaces in three dimensions through the interpolation
 of regularly spaced 2D seismic sections, constrained by wells data and surface geology of the Qaidam basin to reconstruct Cenozoic tectonic history of the Qaidam basin and decipher how the Tibetan plateau was formed. This study presents the subsurface data in conjunction with observations and analysis of the stratigraphic and sedimentary evolution. The Cenozoic deformation history of the Qaidam basin shows geologic synchroneity with uplifting history of the Tibet Plateau. It is therefore proposed that the deformation and uplifting in the south and north edges of the Tibet Plateau was almost synchronous. The total shortening and shortening rate during Cenozoic reached 25.5 km and 11.2% respectively across the Qaidam basin, indicating that the loss of the left-lateral strike slip rates of the Altyn Tagh fault has been structurally transformed into local crustal thickening across NW-trending folds and thrust faults. Meanwhile there is an about 10° vertical component along the strike-slip Altyn Tagh fault, the block oblique slip shows one more growth mechanism of the northeast Tibet.

5

20

### 1 Introduction

The high elevation and great crustal thickness of the Tibet plateau are generally assumed to be resulted from the India–Asia collision, but the processes that have led to the state observed today remain unclear(Liu et al., 2014; Royden et al., 1997; Tapponnier et al., 2001; Wang et al., 2001; Zhang et al., 2013). The Qaidam basin is located in a complex system of compressive structures in the northeast Tibet, and is the largest topographic depression inside the Tibetan plateau (Fig. 1). The tectonic evolution of the Qaidam basin has important implications for unraveling the formation mechanism and growth history of the Tibetan plateau(Burchfiel et al., 1989; Meng and Fang, 2008; Meyer et al., 1998; Wang et al., 2006; Xia et al., 2001; Yin et al., 2008a; Yin et al., 2007; Zheng et al., 2013; Zhou et al., 2006; Meng, 2009). Although much of its late Cenozoic deformation is explained by the collision and subsequent penetration

- 10 of India into Eurasia, how Eurasia deforms in response to the collision is still subject to debate. Most of the papers in fact analyzed just single sectors of the basin, whereas few works dealt with the correlation of each thrust fronts along strike. There are in fact unsolved aspects in terms of geometry, displacements and deformation chronology that can be highlighted by taking into account a bigger area of investigation and by the 3D reconstruction to provide new insights into the kinematics of Eurasia.
- 15 The development of software for 3D modeling has opened a new frontier in the earth sciences, leading to a more accurate spatial analysis of geological structure and to 3D models. Integrating geophysical and geological data, from seismic available database and by geological maps, is possible to define geometrical and geological constraints in order to create 3D surfaces, closed volumes and grids from the constructed objects.

Displacement on the strike-slip faults is absorbed by extension or compression occurring at the termination of the faults, so strike-slip displacement is roughly equal to and in the same direction as shortening or extension. We propose that left slip on the Altyn Tagh fault zone in northern Tibet is similarly absorbed by shortening southeast of the fault

zone within the Qaidam basin, the Qilian mountain and the Kunlun mountain.

#### 2 Methods

The data used in the study include well logs, checkshot data, 2D seismic sections and base map of the study area, and all data were imported into the Landmark's OpenWorks data management environment. Using SynTool software the time interpretation with wells via a one-dimensional synthetic seismogram was tied to match seismic time data with geologic depth data. SeisWorks software was used to interpret  $T_1$ - $T_6$  horizons and faults on seismic sections. Because the seismic data were acquired and processed by different teams, leading to the appearance of misties, we applied the Interactive Seismic Balance tools to analyze and resolve misties in time, phase, and amplitude. After the interpretation, horizons and faults were converted from time to depth domain using TDQ software, and were exported out from OpenWorks for structural modeling.

- Then the data were loaded into the Petrel software to define the faults in the geological model that form the basis for generating the 3D grid. These faults define breaks in the grid, lines along which the horizons inserted later may be offset. The offset which occurs is entirely dependent upon the input data. The Make horizons process generates independent geological horizons from input data. To generate additional horizons using relative distance to existing horizons use the Make zones process. These two processes are used to create the geological zones within the model.
  Layering inserts the fine scale grid cells which will describe vertical variation within each geological zone.
- The section is sequentially restored back in time to its initial stage before the deposition of Cenozoic strata using the computer section balancing software Move(Bigi et al., 2013). The section is divided into discrete blocks bounded by faults and bedding planes, which are then individually manipulated using different restoration algorithms. The flexural slip unfolding algorithm was applied to restore folds. The algorithm works by rotating the limbs of a fold to a datum, an assumed regional, or template geometry. Layer parallel shear is then applied to the rotated fold limbs in order to remove the effects of the flexural slip component of folding. Unfolding occurs about a pin line and points along the pin are not translated. The pin should correspond to the axial surface of the fold. In the flexural slip algorithm, only slip between bedding planes was permitted so that only the area of cross-section and the length of the beds parallel to the chosen reference surface were preserved.

The fault parallel flow algorithm is used to restore faults and is based on particulate laminar flow over a fault ramp. The fault plane is divided into discrete dip domains where a change in the fault's dip is marked by a dip bisector. Flow lines are constructed by connecting points on different dip bisectors of equal distance from the fault plane. Particles in the hanging wall translate along the flow lines, which are parallel to the fault plane, by a defined distance.

5 Decompaction is an essential step to obtain an accurate geometry of the reconstructed structures. Because the lithology of most of the stratigraphic units in Qaidam basin commonly has complex lateral variations, the decompaction has not been taken into account in the section restoration(Zhou et al., 2006).

#### 3 Results

Using the remote sensing satellite data analysis and interpretation, we can identify the distinct structural features including fold, fault, and lineament (Fig. 2, 3). The measurements denoted a good agreement with field data, confirming the potentiality of satellite analyses in geological studies. As a major boundary fault between two large geomorphic units, the Tibet and the Tarim basin, the Altyn Tagh fault displays a remarkable topographic contrast from the northern Plateau to the southern Tarim basin with a vertical throw up to 3500 m(Xu, 2005). The fault cuts various geomorphic surfaces, such as Quaternary fluvial fans and terraces, has developed distinct linear traces along the fault. The axial traces of Qaidam folds to trend NW-SE, Many Qaidam anticlines are fault-propagation folds with steeper dipping fore-limbs and shallower dipping back-limbs. The fore-limbs are located on the northern sides of the axial traces, indicating that the anticlines are propagating northward to northeastward, and the fold geometries may be controlled by reactivation or inversion of preexisting south to southwest-dipping faults in the underlying basement(Kapp et al., 2011).

The Altyn Tagh fault shows distinct linear feature on remote sensing images and field observations (Fig. 3). The offset gully and vertical fault plane indicates it is a left-lateral strike-slip fault (Fig. 3b, 3d), meanwhile the fault scarp and 10 degree oblique striations on the fault plane means that the fault has vertical component (Fig. 3c, 3e).

3D modeling allows to detect and analyze complex spatial relationships, leading to a better characterization of both

5

exposed and subsurface geology(Caumon et al., 2009; Escalona and Mann, 2003; Valcarce et al., 2006). It is especially useful in cases where the structures are not cylindrical and 2D visualization may be not completely useful. The increasing ability of geological software packages to integrate and visualize different datasets in a 3D framework allows for a more critical evaluation and triggers a convergence of interpretations. The geological and structural setting of the Qaidam basin has been reconstructed in a 3D model through the interpolation of regularly spaced 2D seismic sections, constrained by wells data and surface geology (Figs. 2 and 4).

- In this area, the main geological structure is that the basin is bounded by the southwest-directed Kunlun thrust belt in the west and the northeast-directed Qilian mountain thrust belt in the east (Fig. 4). The thrusts terminating against the Altyn Tagh fault and the lateral ramps are subparallel to the left-slip Altyn Tagh fault and their development may result
- 10 from distributed left-slip deformation that transfers motion from the Altyn Tagh fault to the left-slip ramps via the linking thrusts. The Qaidam basin began to subside due to crustal shortening in the Eocene, and it has subsequently evolved into an independent basin since the Miocene. The main depocenter was noticeably persistent in the middle of the western Qaidam basin from Eocene to Miocene time, and then it shifted to the east. The Qaidam basin was generated as a result of crustal buckling or folding, manifesting itself as a synclinal depression. The crustal folding 15 model can account for a number of observations, including localization of the depocenter in the middle of the basin, nearly concomitant deformation on the southern and northern sides of the Qaidam basin, occurrence of major high-angle reverse faults at basin margins. The contraction between the upper and lower crust has been decoupled, then lower crustal flow and thermal events in the upper mantle could be additional causes of plateau uplift across the Kunlun mountain, Qaidam basin, and Qilian mountain.
- The two-dimensional time-depth converted seismic profile AB was used for the construction of the geological section, which is perpendicular to the trends of structures. The restoration results of the geological section AB (Fig. 6), including total shortening, average shortening rates, and shortening rates during different time periods, are listed in Table 1. The total shortening and shortening ratio during Cenozoic respectively reaches 25.5 km and 11.2% across the Qaidam basin, which is consistent with previous works(Wang et al., 2011; Yin et al., 2008b; Zhou et al., 2006). 42.9%
- and 34% of total shortening took place in the periods since 5.1 Ma and since 2.8 Ma, respectively. The shortening rates

5

from the restoration of the section indicate that the Qaidam basin has been undergoing continuous shortening since the beginning of Cenozoic with two relatively fast shortening phases, the first during  $E_3^2$  layer (42.8–40.5 Ma) and the second during Q layer (2.8 Ma–present), and the shortening rates reached the maximum values to 3.09 mm/yr since 2.8 Ma, whereas the average rate during the Cenozoic is 0.39 mm/yr (Tab. 1). From 65 Ma to 42.8 Ma, the shortening rate increased from 0.14 to 1.08 mm/yr, then from 40.5 Ma to 12 Ma, the rate decreased to 0.09 mm/yr, and finally the rate increased to the maximum in Quaternary.

4 Discussion

There are two models of how the high elevations in Tibet formed. One is continuous thickening and widespread viscous flow of the crust and mantle of the entire plateau, and the other is time-dependent, localized shear between coherent lithospheric blocks(Clark, 2012; Métivier et al., 1998; Tapponnier et al., 2001; Thatcher, 2007; Zhang et al., 2013). In this study, the Cenozoic deformation history of the Qaidam Basin (Fig. 6) shows that the basin experienced continuous compression since the beginning of Cenozoic. It Means that the deformation and uplifting in the south and north edges of the Tibet Plateau were almost synchronous. The restoration shows that the deformation of Qaidam basin was not uniform(Zhuang et al., 2011), and it experienced an increase and a decrease, then an increase process. In later Eocene and Quaternary, the basin has two relatively fast shortening phases, and from later Miocene to present, the rate has been increasing to the maximum value ever, indicating the northeastern Tibet will keep strong deformation in the future.

The Altyn Tagh fault is the major boundary fault on the northern margin of Tibet, and the left slip on the fault zone is related to and absorbed by crustal shortening within the Qilian mountain, the Qaidam basin, and other convergent structures south of the Altyn Tagh fault zone(Burchfiel et al., 1989). Slip vector analyses also indicate that the loss of the sinistral slip rates from west to east has structurally transformed into local crustal shortening perpendicular to the active thrust faults and strong uplifting of the thrust sheets to form the NW-trending Qilian moutain(Xu, 2005). Correspondingly, the Cenozoic deformation history of the Qaidam Basin has implications for the evolution of Altyn

Tagh fault. The total strike-slip offset of the Altyn Tagh fault across the Qaidam basin is about 25.5 km during Cenozoic. From west to east of the Qaidam basin, the slip rate of the Altyn Tagh fault decreased at about 3 mm/yr in Quaternary. The thrusts and folds developed in Qaidam basin absorbed the movement of the Altyn Tagh fault. In the field work, we observed that the strike-slip Altyn Tagh fault has about 10-degree vertical component (Fig. 5e), which is supported by the uplifted young fault scarp at the south side of the fault on the flood plain (Fig. 5c).

- The crust-buckling model(Meng and Fang, 2008; Meng, 2009) gives an explanation of the initiation and development of the Qaidam basin. The model requires the low crustal of eastern Tibet to be weak enough to flow outward from the interior of the Tibet Plateau(Wang et al., 2012). The outward growth of eastern Tibet is probably driven by lower crustal flow under the lateral pressure gradients. Since the upper crust of the blocks is dominated by brittle deformation,
- the ductile flow of the lower crust would drag the brittle upper crustal blocks to move with respect to each other. The interactions among the brittle upper crustal blocks cause strain accumulations among their bounding faults to generate large earthquakes(Zhang, 2013). A recent study determined both the P- and S-wave velocity structure across the central Qaidam basin along a 350-km-long seismic refraction/wide-angle reflection profile extending from the Kunlun mountain through the central Qaidam basin into the Qilian mountain, and shows brittle deformation in the upper crust through thrust folding, and ductile deformation of the lower crust though pure shear deformation(Zhao et al., 2013). The Qaidam basin shows overall a synclinorium, with a narrow NW-SE direction distribution, and on the east and west
- side of the basin, main piedmont thrusts control the boundary, and the strike-slip Altyn Tagh fault limits the north side of the basin (Fig. 5). The synclinorium shape supports that the upper crust of Qaidam basin has been compressed to a depression, under the flow of lower crust.
- The Altyn Tagh fault as a boundary and transform fault plays an important role in crustal shortening and growth of the northeast Tibet. The folding and thrusting on the south side of the Altyn Tagh fault lead to the shortening and thickening of the upper crust, meanwhile the south block has a whole oblique uplift along the fault (Fig. 7). The driving force may come from the southward subduction of lithospheric mantle overlying detached crust(Meng, 2009).

Acknowledgements. This study is financially supported by the Important Direction Project of Knowledge Innovation in

Resource and Environment Field, Chinese Academy of Sciences (KZCX2-EW-QN112), and the fund from the Key Laboratory of Petroleum Resources, Gansu Province, Institute of Geology and Geophysics, Chinese Academy of Sciences. We thank Global Land Cover Facility (GLCF) for providing the DEM data set (<u>http://glcf.umd.edu/</u>).

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

25

## **Table captions**

| Layer                              | Length (mm) | Age (yr) | Shortening Length<br>(mm) | Time Interval<br>(yr) | Percentage of total shortening (%) | Shortening<br>Rate(mm/yr) |
|------------------------------------|-------------|----------|---------------------------|-----------------------|------------------------------------|---------------------------|
| Present                            | 201802000   | 0        | 8652000                   | 2800000               | 34.0                               | 3.09                      |
| Q (T <sub>0</sub> )                | 210454000   | 2800000  | 2274000                   | 2300000               | 8.9                                | 0.99                      |
| N2 <sup>3</sup> (T1)               | 212728000   | 5100000  | 4060000                   | 6900000               | 15.9                               | 0.59                      |
| $N_2^2 (T_2)$                      | 216788000   | 12000000 | 1150000                   | 12600000              | 4.5                                | 0.09                      |
| N2 <sup>1</sup> (T2)               | 217938000   | 24600000 | 3698000                   | 15900000              | 14.5                               | 0.23                      |
| N1 (T3)                            | 221636000   | 40500000 | 2478000                   | 2300000               | 9.7                                | 1.08                      |
| E3 <sup>2</sup> (T4)               | 224114000   | 42800000 | 1457000                   | 9700000               | 5.7                                | 0.15                      |
| E3 <sup>1</sup> (T5)               | 225571000   | 52500000 | 1722000                   | 12500000              | 6.8                                | 0.14                      |
| E <sub>1+2</sub> (T <sub>R</sub> ) | 227293000   | 65000000 |                           |                       |                                    |                           |
| Total                              | 227293000   |          | 25491000                  | 65000000              | 11.2                               | 0.39                      |

Table 1. Restoration measurements and shortening rates of section AB

## **Figure captions**