# Peer review of "Three-dimensional structural model of the Qaidam basin: Implications for crustal shortening and growth of the northeast Tibet"

_Solid Earth, 2016_

## Referee Comment (RC1) · G. Vignaroli (Referee) · 8 Jun 2016

Comments on the manuscript:

**"Three-dimensional structural model of the Qaidam basin: Implications for crustal shortening and growth of the northeast Tibet"**

By J. Guo et al.

**Submitted to Solid Earth (se-2016-74)**

This manuscript deals with the structural evolution of the Qaidam basin (northeast Tibet) by using geological and geophysical datasets. The main geological surfaces (both stratigraphic and tectonic) are reconstructed in 3D structural modelling, providing a sequential restoration of the whole basin through time. The results are used to infer the tectonic role of the Qaidam basin within the Tibet plateau development.

Taking into account the aims and the implications, this work is of interest for an international audience and it is appropriate for being published in *Solid Earth*. Anyway, the present version of the manuscript suffers of an inadequate organisation of the text and insufficient data presentation. I recommend major revisions and my comments/suggestions follow here.

**Major comments**

Abstract. Some information are not clearly stated. For a reader that has not familiarity with the geology of Tibet, the relationships between the Tibet plateau, the Qaidam basin and the Altyn Tagh fault are obscure. Briefly, you should explain, starting from the Abstract, how the Qaidam basin, and its relationships with the Altyn Tagh fault, are crucial for understanding the tectonic evolution of the Tibet region.

Introduction. The last part of this paragraph (e.g., page 6, lines 8-12) should be rewritten. In particular, here it should be clearly stated which are: (i) the main objectives of the work, (ii) the main obtained results, and (iii) which are the tectonic implications derived. I invite you to clearly state which is the novelty of your work with respect to that already published by other researchers (I suggest stressing on the 3D reconstruction derived from integration of geological and geophysical datasets).

Regional tectonic context (NEW PARAGRAPH). I propose to include this new paragraph in order to clarify (i) the geological context, (ii) the models proposed for Cenozoic deformation in the Qaidam basin, (iii) the processes advocated as responsible for the present-day structural architecture of the Qaidam basin, and (iv) the structural/tectonic relationships between the Qaidam basin and the surrounding domains (i.e., the Altyn Tagh fault, the Kunlum Mountains, the Qiliam Mountains). I think this is important for providing a clear background of your study area. You can use some parts of the main text that are presently dispersed within the Introduction paragraph (page 2, lines 22-25), the Results paragraph (page 5, line 20 to page 6, line 5), the Discussion paragraph (page 6, lines 22-25; page 7, lines 5-11).

By providing such a background, it will be more clear which is your innovative contribution in defining the tectonic evolution of the Qaidam basin with respect to already published works (for example, in comparison to Yin et al., 2008, GSA Bulletin).

Results. This paragraph should include and detail your dataset. Indeed, it does not provide enough information on the dataset used and how these data have been integrated for producing the 3D restoration.

I suggest to re-organise this paragraph by adding few specific sub-paragraphs. For example:

- Structural dataset: here you could focus the attention on the structural elements characterising the Qaidam basin (such as folds, faults and lineaments) and describe their properties (distribution, persistence, attitude, crosscut relationships, etc...). This may include your text in page 4 (lines 24-29) and page 5 (lines 1-11). Here, I would like to see a more detailed characterisation of the Altyn Tagh fault in terms of fault architecture (thickness and length of damage zone and fault core) and structural data. For the latter, I suggest to provide more evidence of fault kinematics by using field pictures, line drawings, stereographic projections and statistical analysis of the structural elements (e.g., fault strike, pitch values);
- Wells dataset: a detailed description of the well stratigraphies completely lacks in this work (or did I miss something in downloading the manuscript?) I think you can not exclude this dataset that you used for constraining the seismic profiles and, then, the 3D restoration.
- Geophysical dataset: here you could present the seismic profile you used by describing geometries and thicknesses of layers, as well as geometries and offsets of the main faults.

Discussion. I think this paragraph could be improved in the light of text re-organisation within the previous paragraphs. In particular, I propose to better explain why and how your restoration is different from that already published (e.g., Yin et al., 2008; Zhou et al., 2006).

Conclusions (NEW PARAGRAPH). I encourage you to introduce a conclusive paragraph in which you can stress on your main results and implications (at the scale of the Qaidam basin and at the scale of the Tibet plateau).

**Minor Comments**

- Page 2, lines 1-4: too long phrase. Please, split it into two concise phrases.
- Page 2, line 3: add "(northeast Tibet)" after Qaidam basin.
- Page 3, lines 8-10. I don't understand the relations between this statement and the previous one. I suggest to delete it (or remove and to use it within the Discussion).
- Page 3, lines 11-12: Qiliam and Kunlun mountains have been never mentioned before. Not easy to understand their relationships with the Qaidam basin.
- Pages 4-5, lines 22-29 and 1-7: all the information should be better supported by a more detailed geological-structural map in Figure 1 (see below) and by field pictures in Figure 2.
- Page 5, line 2: delete "to" and change the comma in full stop.
- Page 5, lines 12-19: this is no part of the Results paragraph. It should be moved within Methodology and re-arranged.
- Page 5, line 22: "(*Fig. 4*)" not appropriate to cite there. Remove.
- Page 5, lines 22-25: this phrase is very confusing. I suggest to rewrite it. Anyway, as you are declaring that thrusts are related to the Altyn Tagh fault, I would like to understand if you consider them either (i) fault termination in a horsetail arrangement, or (ii) contractional

duplex in restraining bend of the Altyn Tagh fault, or (iii) resulting from restraining stepover between two parallel faults (i.e., the Altyn Tagh fault and the Kunlun fault).

- Page 5, line 20 to page 6, line 5: it seems to me that the information derives from previous works. If so, I strongly suggest to delete it from here and to merge it within the paragraph concerning the Regional Tectonic Context.
- Page 6, line 8: why do you introduce Fig. 6 before Fig. 5?
- Page 6, lines 22-25: delete this information and merge it within the Regional Tectonic Context.
- Page 6, line 27: "*Means*" with lowercase letter.
- Page 7, line 3: change "indicating" with "suggesting".
- Page 7, lines 5-10: this is part of the Regional Tectonic Context.
- Figure 1. I suggest these caprions:
  - (A): Structural map of the Tibet Plateau within the Eurasia/India continental collision. Black arrow indicates present-day displacement of India continental block.
  - (B): Shaded relief map......
- Figure 1: provide latitude degrees in the shaded relief map. Provide a legend for explaining the meaning of A-B, C-D, line with triangles (thrust).
- Figure 2:
- indicate orientation and an approximate scale for all pictures.
- provide location of these structures in Figure 1.
- Improve legibility by introducing some labels and symbols indicating bedding (strike and dip), trace of possible fault segments, offsets along fault segments, ....
- Within the caption: change "*filed*" with "*field*".
- Figure 3:
- Add scale for all figures.
- In printed page, labels and symbols are not legible. Improve them by changing colour and/or increasing the size.
- No caption is provided for (A).
- In (B): which is the meaning of the big red arrow and the black one? Please, indicate the river you are considering.
- In (E): the stereographic projection is not visible. As suggested above, please provide more accurate structural dataset.
- Within the caption: change "filed" with "field".
- The caption of this figure should be re-organised by following the figure sequence from (A) to (F).
- Statement in lines 5-6 makes no sense if you don't provide an accurate structural dataset and statistical analysis of the pitch values.
- Figure 3:
- This figure is not explained within the main text. Geometries, depths and offsets of the main seismic reflectors should be described.

- All red lines are thrusts? Please provide a legend.
- Indicate the Altyn Tagh fault in seismic profile C-D
- Figure 5:
- Which is the meaning of T0, T21, T4, and TR as reported in caption? Are these time steps? Depths? Please, indicate within the figures.
- What does the green arrow? North? India displacement?
- Figure 6: following you model, uplift attained only after N23 layer deposition. Indeed, starting from the Abstract, you stated that deformation and uplift were synchronous in Tibet (thus in the Qaidam basin). Please, explain.
- Figure 7: what does the green symbol? Does arrow point the north?

I hope my comments and suggestions can improve the manuscript.

My name may be communicated to the authors. Sincerely, Gianluca Vignaroli

Istituto di Geologia Ambientale e Geoingegneria (IGAG) Consiglio Nazionale delle Ricerche (CNR) Area della Ricerca di Roma 1, Via Salaria Km 29, 300–00015 Monterotondo Stazione, Rome, Italy gianluca.vignaroli@igaq.cnr.it

---

## Referee Comment (RC2) · Anonymous Referee #2 · 26 Jun 2016

"Three-dimensional structural model of the Qaidam basin: Implications for crustal shortening and growth of the northeast Tibet" by Guo, et al.

This paper deals with the geological evolution of a basin in the Tibetan Plateau. The theme is very interesting overall because it deals with the geological evolution of an interesting and very complex area. The methodology is innovative and the used tools are surely useful. Nevertheless, the paper lacks of data. These are poor or not presented in an exhaustive way. For example, the geometry and kinematics of the faults have not been well documented, as well as the stratigraphy of the basin. In fact, the relationships between structures and sedimentation are primary analyses to be done dealing with evolution of a basin. This is completely omitted in the paper. Also the

seismic interpretation needs of more strong geological constrains. This is why, in my opinion, this paper has to be improved a lot before to be published. Best regards

---

## Author Comment (AC1) · 20 Jul 2016

Dear reviewer:

I am very grateful to your comments for the manuscript. According with your advice, we amended the relevant part in manuscript. Some of your questions were answered below.

Comment:

This paper deals with the geological evolution of a basin in the Tibetan Plateau. The theme is very interesting overall because it deals with the geological evolution of an interesting and very complex area. The methodology is innovative and the used tools

are surely useful. Nevertheless, the paper lacks of data. These are poor or not presented in an exhaustive way. For example, the geometry and kinematics of the faults have not been well documented, as well as the stratigraphy of the basin. In fact, the relationships between structures and sedimentation are primary analyses to be done dealing with evolution of a basin. This is completely omitted in the paper. Also the seismic interpretation needs of more strong geological constrains. This is why, in my opinion, this paper has to be improved a lot before to be published.

Answer: In the revision, we have re-organized the manuscript and added some description. More about the Altyn Tagh fault:

The Altyn Tagh fault extends for at least 1,500 km from the west Kunlun thrust zone in the southwest to the edge of the Qilian mountains in the northeast. It is divided into three main sections: southwestern, central and northeastern. There is one major splay fault, the north Altyn fault. The main active fault trace of the fault lies within a zone of secondary structures that is about 100 km wide in the central section; The horizontally offset gully, which have been displaced from several meters to several hundred meters, and vertical fault plane, which has an 82° dip and 76° strike, indicates it is a left-lateral strike-slip fault (Fig. 3b, 3d), meanwhile the uplifted fault scarp and 11° oblique striations on the fault plane means that the fault has a vertical component (Fig. 3c, 3e). The thickness of the fault core is over several hundred meters on the outcrop (Fig. 3d). Faults and folds extending northward are cut and terminated by the Altyn Tagh fault.

we added a paragraph to describe the stratigraphy:

Cenozoic strata in the Qaidam basin, over 15 km in total thickness, contains a continuous sequence of lacustrine sediments, and the depositional center of the basin shifted progressively towards the east during Cenozoic times. The shift was accompanied by uplift in the west and subsidence in the east of the basin. Cenozoic stratigraphic division and age assignments across Qaidam basin are based on outcrop geology and

its correlation with subsurface well data. The units are: Paleocene to early Eocene of the Lulehe formation (65–52.5 Ma); middle Eocene of the lower Ganchaigou formation (52.5–42.8 Ma); late Eocene of the lower Ganchaigou formation (42.8–40.5 Ma); late Eocene to Oligocene of the upper Ganchaigou formation (40.5–24.6 Ma); early to middle Miocene of the lower Youshashan formation (24.6–12 Ma); late Miocene of the upper Youshashan formation (12–5.1 Ma); Pliocene of the Shizigou formation (5.1–2.8 Ma); and late Pliocene to Quaternary of the Qigequan formation (2.8 Ma–present) (Zhou et al., 2006). Paleocene and Eocene strata has a few small outcrops in northern Qaidam but are not widespread in the rest of the basin. Oligocene strata are more widespread and include lacustrine rocks within the lower Ganchaigou formation in north-western Qaidam. Lacustrine deposition was also widespread in Qaidam during the Miocene, whereas lesser amounts of lacustrine strata are present in lower Pliocene sections. Upper Pliocene and Quaternary sections are generally coarser grained fluvial and alluvial deposits, which are quite thick and reflect high sedimentation rates. Where thickest, Oligocene–Miocene strata exceed 3 km, whereas Pliocene and Quaternary sections in places exceed 5.5 and 2.8 km, respectively (Hanson et al., 2001).

About the seismic interpretation:

A total of 17 seismic lines were gathered, and the total length is over 3400 km, from a single length of 80 km to 450 km. Two-way time of seismic reflection is 6 seconds (Fig. 4), and seismic lines are parallel and perpendicular respectively to the structural direction. Seismic project datum is 2772 m.

A total of 9 wells were used to time-depth conversion, and these wells are located in the vicinity of seismic lines. The depth of wells ranges from 3000 to 6000 m. Well logging data are mainly sonic logs, density logs, and Geological strata data.

The synthetic seismograms were used for the seismic and geologic horizon calibration. Combined with drilling data, the geological horizon system of the basin was established, and the seismic reflection characteristics of each layer was basically defined.

T6 is the bottom of Mesozoic (M) and the top reflection of bedrock. It is an unconformity surface and the low frequency is continuous, when the waveform changes strongly in the west of the basin. Whereas it is clear and reliable in the east of the basin. TR is the bottom reflection of E1+2. It is major angular unconformity in the basin, and is a phase axis under 2-3 continuous strong phases. Due to distinct characteristics, TR is a standard reflection in the basin.

T5 is the bottom reflection of E31. It is characterized by the high amplitude, continuous features.

T4 is the bottom reflection of E32. It is a standard reflection in the basin. It is characterized by 2 phase, strong amplitude and continuous reflection under a set of blank reflection in the western basin. In the eastern basin, it lies at the top of a group strong reflection.

T3 is the bottom reflection of N1. It shows medium-high amplitude and continuous reflection. In the west of the basin, the wave energy becomes weak.

T2 is the bottom reflection of N21. It is a 2 phase, strong amplitude and continuous reflection. The variation of amplitude becomes large and continuity becomes poor in the eastern margin of the basin.

T2ËŁ is the bottom reflection of N22. It is a regional unconformity, and reflection is continuous except margin and eroded area.

T1 is the bottom reflection of N23. In the depression, it has good continuity, whereas it becomes unrest at the margin.

T0 is the bottom reflection of Q. It shows 1-2 phase, medium-high amplitude, and continuous features in the depression, whereas the reflection is not stable at the margin, and at some area it is missing due to the erosion.

After completing seismic and geologic horizon calibration, the results were taken into the seismic profiles to establish the interpretation network. Through repeated comparison, repeated modification of the drilling and seismic data, the seismic and geological layer system of the whole basin was built up.

I hope that my response can clarify your concern.

Sincerely,

Jianming Guo

Key Laboratory of Petroleum Resources, Gansu Province / Key Laboratory of Petroleum Resources Research Institute of Geology and Geophysics, Chinese Academy of Sciences Lanzhou 730000, China.

gjm2001cn@yahoo.com

---

## Author Comment (AC2) · 20 Jul 2016

Dear Dr. Vignaroli:

We are very grateful to your thoughtful and constructive comments for the manuscript. According with your advice, we amended the relevant part in manuscript. Some of your questions were answered below.

Major comments

Abstract. Some information are not clearly stated. For a reader that has not familiarity with the geology of Tibet, the relationships between the Tibet plateau, the Qaidam basin and the Altyn Tagh fault are obscure. Briefly, you should explain, starting from

the Abstract, how the Qaidam basin, and its relationships with the Altyn Tagh fault, are crucial for understanding the tectonic evolution of the Tibet region.

Answer. In the Abstract, we added following information to give information on the relationships in geological blocks in the manuscript: The Qaidam basin, bounded by the Altyn Tagh fault in the north, is located in the northeast of the Tibet plateau, and it has important implications for understanding the history and mechanism of Tibetan plateau formation during the Cenozoic Indo-Asian collision.

Introduction. The last part of this paragraph (e.g., page 6, lines 8-12) should be rewritten. In particular, here it should be clearly stated which are: (i) the main objectives of the work, (ii) the main obtained results, and (iii) which are the tectonic implications derived. I invite you to clearly state which is the novelty of your work with respect to that already published by other researchers (I suggest stressing on the 3D reconstruction derived from integration of geological and geophysical datasets).

Answer. We rewritten the last part of Introduction: There exist different views on the structure and evolution of the Qaidam Basin, and the corresponding tectonic evolution models has been established (Meng, 2009). The first model is the extension-compression two-stage model, and extensional tectonics occurred in the late Cretaceous - early Tertiary, while squeezing construction began in late Oligocene (Xia et al., 2001). The second is the flexure subsidence model of foreland basin, that at present is generally accepted, the main reason is that Qaidam basin is currently defined by thrust faults at northeast and southwest sides (Métivier et al., 1998). The third is the piggy-back basin model, and Qaidam upper-crustal structures can be explained by thrusting along a mid-crustal décollement (Yin et al., 2008b). The fourth is the crust-buckling model, which is generated under horizontal compression (Meng and Fang, 2008). The main models were generally studied by using 2D structures, in this study, we used the 3D method to establish a complete structure to reveal the spatial distribution of faults and strata of the basin, and the result indicates that the crust-buckling model is more consistent with 3d modeling in the uplift of northeastern Tibet. With geologic section

restoration, we measured the total shortening and shortening ratio during Cenozoic. It shows that the deformation process of Qaidam basin was not uniform.

Regional tectonic context (NEW PARAGRAPH). I propose to include this new paragraph in order to clarify (i) the geological context, (ii) the models proposed for Cenozoic deformation in the Qaidam basin, (iii) the processes advocated as responsible for the present-day structural architecture of the Qaidam basin, and (iv) the structural/tectonic relationships between the Qaidam basin and the surrounding domains (i.e., the Altyn Tagh fault, the Kunlum Mountains, the Qiliam Mountains). I think this is important for providing a clear background of your study area. You can use some parts of the main text that are presently dispersed within the Introduction paragraph (page 2, lines 2225), the Results paragraph (page 5, line 20 to page 6, line 5), the Discussion paragraph (page 6, lines 22-25; page 7, lines 5-11). By providing such a background, it will be more clear which is your innovative contribution in defining the tectonic evolution of the Qaidam basin with respect to already published works (for example, in comparison to Yin et al., 2008, GSA Bulletin).

Answer. A new paragraph of Regional geology was added: The Qaidam Basin has a rhombic shape that is located on the north-eastern margin of the Tibetan plateau. The basin covers an area of approximately $1.2 \times 105$ km2 and has elevations of 2500–3000 m (Fig. 1). The deformation of major faults surrounding the basin and the Cenozoic strata within the basin provides a record of the basin–mountain tectonic framework and geodynamic setting. Specifically, four major faults control the evolution of the basin: the Kunlun fault to the south, the Altyn Tagh fault to the northwest, the Kunlun mountain thrust belt on the southern margin, and the Qilian mountain thrust belt to the northeast of the basin. The basin is divided into two contrasting sectors, and the the northwest sector experienced significantly stronger deformation than the southeast sector (Fig. 1), containing a series of NW–SE trending thrust-fold belts.

Cenozoic strata in the Qaidam basin, over 15 km in total thickness, contains a continuous sequence of lacustrine sediments, and the depositional centre of the basin shifted
progressively towards the east during Cenozoic times. The shift was accompanied by uplift in the west and subsidence in the east of the basin. Cenozoic stratigraphic division and age assignments across Qaidam basin are based on outcrop geology and its correlation with subsurface data. The units are: Paleocene to early Eocene of the Lulehe formation (65–52.5 Ma); middle Eocene of the lower Ganchaigou formation (52.5–42.8 Ma); late Eocene of the lower Ganchaigou formation (42.8–40.5 Ma); late Eocene to Oligocene of the upper Ganchaigou formation (40.5–24.6 Ma); early to middle Miocene of the lower Youshashan formation (24.6–12 Ma); late Miocene of the upper Youshashan formation (12–5.1 Ma); Pliocene of the Shizigou formation (5.1–2.8 Ma); and late Pliocene to Quaternary of the Qigequan formation (2.8 Ma–present) (Zhou et al., 2006). Paleocene and Eocene strata has a few small outcrops in northern Qaidam but are not widespread in the rest of the basin. Oligocene strata are more widespread and include lacustrine rocks within the lower Ganchaigou formation in north-western Qaidam. Lacustrine deposition was also widespread in Qaidam during the Miocene, whereas lesser amounts of lacustrine strata are present in lower Pliocene sections. Upper Pliocene and Quaternary sections are generally coarser grained fluvial and alluvial deposits, which are quite thick and reflect high sedimentation rates. Where thickest, Oligocene–Miocene strata exceed 3 km, whereas Pliocene and Quaternary sections in places exceed 5.5 and 2.8 km, respectively (Hanson et al., 2001). The Altyn Tagh fault is the major boundary fault on the northern margin of Tibet, and the left slip on the fault zone is related to and absorbed by crustal shortening within the Qilian mountain, the Qaidam basin, and other convergent structures south of the Altyn Tagh fault zone (Burchfiel et al., 1989). Slip vector analyses also indicate that the loss of the sinistral slip rates from west to east has structurally transformed into local crustal shortening perpendicular to the active thrust faults and strong uplifting of the thrust sheets to form the NW-trending Qilian mountain (Xu, 2005).

Results. This paragraph should include and detail your dataset. Indeed, it does not provide enough information on the dataset used and how these data have been integrated for producing the 3D restoration. I suggest to re-organise this paragraph by

adding few specific sub-paragraphs. For example: - Structural dataset: here you could focus the attention on the structural elements characterising the Qaidam basin (such as folds, faults and lineaments) and describe their properties (distribution, persistence, attitude, crosscut relationships, etc...). This may include your text in page 4 (lines 24-29) and page 5 (lines 1-11). Here, I would like to see a more detailed characterisation of the Altyn Tagh fault in terms of fault architecture (thickness and length of damage zone and fault core) and structural data. For the latter, I suggest to provide more evidence of fault kinematics by using field pictures, line drawings, stereographic projections and statistical analysis of the structural elements (e.g., fault strike, pitch values); - Wells dataset: a detailed description of the well stratigraphies completely lacks in this work (or did I miss something in downloading the manuscript?) I think you can not exclude this dataset that you used for constraining the seismic profiles and, then, the 3D restoration. - Geophysical dataset: here you could present the seismic profile you used by describing geometries and thicknesses of layers, as well as geometries and offsets of the main faults.

Answer. In Methods, we added Datasets to provide remote sensing, seismic line and well data information: The typical input data for a 3D structural modeling project can be quite diverse and may include field observations (for instance, stratigraphic contacts and orientations, fault planes), interpretive maps and cross-sections, remote sensing pictures, and reflection seismic, isochrones maps, seismic profiles and borehole data. Each data type has its specific features, which will act upon how it is integrated in the modeling process and affect the quality of the model (Caumon et al., 2009). In this study, we collected SRTM DEM, Landsat ETM and QuickBird remote sensing data to identify faults and folds on the surface (Figs 1, 2 and 3). The Shuttle Radar Topography Mission (SRTM) is an international research effort that obtained digital elevation models (DEM) on a near-global scale to generate the most complete digital topographic database of Earth based on the spaceborne interferometric synthetic aperture radar, and the ground resolution of one pixel is 90 m. The Landsat Enhanced Thematic Mapper (ETM) data cover the visible, near-infrared, shortwave, and thermal infrared

spectral bands of the electromagnetic spectrum. The panchromatic resolution is 15 m, and multispectral resolution is 30 m. QuickBird was a high-resolution commercial earth observation satellite, and its panchromatic resolution is 0.6 m, multispectral resolution is 2.4 m. DEM and ETM data covering the Qaidam basin were downloaded freely from Global Land Cover Facility (http://glcf.umd.edu/). The linear features of fault and fold in remote sensing images are clear, according to the results of remote sensing image interpretation, in the field we selected a number of representative points to observe distribution of faults and folds (Figs. 2, 3).

A total of 17 seismic lines were gathered, and the total length is over 3400 km, from a single length of 80 kilometers to 450 kilometers. Two-way time of seismic reflection is 6 seconds (Fig. 4), and seismic lines are parallel and perpendicular respectively to the structural direction. Seismic project datum is 2772 meter.

A total of 9 wells were used to time-depth conversion, and these wells are located in the vicinity of seismic lines. The depth of wells ranges from 3000 to 6000 meters. Well logging data are mainly sonic logs, density logs, and Geological strata data.

The synthetic seismograms were used for the seismic and geologic horizon calibration. Combined with drilling data, the geological horizon system of the basin was established, and the seismic reflection characteristics of each layer was basically defined.

T6 is the bottom of Mesozoic (M) and the top reflection of bedrock. It is an unconformity surface and the low frequency is continuous, when the waveform changes strongly in the west of the basin. Whereas it is clear and reliable in the east of the basin.

TR is the bottom reflection of E1+2. It is major angular unconformity in the basin, and is a phase axis under 2-3 continuous strong phases. Due to distinct characteristics, TR is a standard reflection in the basin.

T5 is the bottom reflection of E31. It is characterized by the high amplitude, continuous features.

T4 is the bottom reflection of E32. It is a standard reflection in the basin. It is characterized by 2 phase, strong amplitude and continuous reflection under a set of blank reflection in the western basin. In the eastern basin, it lies at the top of a group strong reflection.

T3 is the bottom reflection of N1. It shows medium-high amplitude and continuous reflection. In the west of the basin, the wave energy becomes weak.

T2 is the bottom reflection of N21. It is a 2 phase, strong amplitude and continuous reflection. The variation of amplitude becomes large and continuity becomes poor in the eastern margin of the basin.

T2ËŁ is the bottom reflection of N22. It is a regional unconformity, and reflection is continuous except margin and eroded area.

T1 is the bottom reflection of N23. In the depression, it has good continuity, whereas it becomes unrest at the margin.

T0 is the bottom reflection of Q. It shows 1-2 phase, medium-high amplitude, and continuous features in the depression, whereas the reflection is not stable at the margin, and at some area it is missing due to the erosion.

After completing seismic and geologic horizon calibration, the results were taken into the seismic profiles to establish the interpretation network. Through repeated comparison, repeated modification of the drilling and seismic data, the seismic and geological layer system of the whole basin was built up.

In Results, we described the Altyn Tagh fault: The Altyn Tagh fault extends for at least 1,500 km from the west Kunlun thrust zone in the southwest to the edge of the Qilian mountains in the northeast. It is divided into three main sections: southwestern, central and northeastern. There is one major splay fault, the north Altyn fault. The main active fault trace of the fault lies within a zone of secondary structures that is about 100 km wide in the central section. The fault cuts various geomorphic surfaces, such as Quaternary fluvial fans and terraces, has developed distinct linear traces along the fault. The Altyn Tagh fault shows distinct linear feature on remote sensing images and field observations (Fig. 3). The horizontally offset gully, which have been displaced from several meters to several hundred meters, and vertical fault plane, which has an 82° dip and 76° strike, indicates it is a left-lateral strike-slip fault (Fig. 3b, 3d), meanwhile the uplifted fault scarp and 11° oblique striations on the fault plane means that the fault has a vertical component (Fig. 3c, 3e). The thickness of the fault core is over several hundred meters on the outcrop (Fig. 3d). Faults and folds extending northward are cut and terminated by the Altyn Tagh fault. The Well stratigraphies are listed in the Regional geology: Cenozoic stratigraphic division and age assignments across Qaidam basin are based on outcrop geology and its correlation with subsurface well data. The units are: Paleocene to early Eocene of the Lulehe formation (65–52.5 Ma); middle Eocene of the lower Ganchaigou formation (52.5–42.8 Ma); late Eocene of the lower Ganchaigou formation (42.8–40.5 Ma); late Eocene to Oligocene of the upper Ganchaigou formation (40.5–24.6 Ma); early to middle Miocene of the lower Youshashan formation (24.6–12 Ma); late Miocene of the upper Youshashan formation (12–5.1 Ma); Pliocene of the Shizigou formation (5.1–2.8 Ma); and late Pliocene to Quaternary of the Qigequan formation (2.8 Ma–present) (Zhou et al., 2006). Paleocene and Eocene strata has a few small outcrops in northern Qaidam but are not widespread in the rest of the basin. Oligocene strata are more widespread and include lacustrine rocks within the lower Ganchaigou formation in north-western Qaidam. Lacustrine deposition was also widespread in Qaidam during the Miocene, whereas lesser amounts of lacustrine strata are present in lower Pliocene sections. Upper Pliocene and Quaternary sections are generally coarser grained fluvial and alluvial deposits, which are quite thick and reflect high sedimentation rates. Where thickest, Oligocene–Miocene strata exceed 3 km, whereas Pliocene and Quaternary sections in places exceed 5.5 and 2.8 km, respectively (Hanson et al., 2001).

Geophysical dataset: The length of the section is over 200 km, and entire section shows a synclinorium shape. Strata on both sides is thin, meanwhile in the middle

is thick, and the thickest part is over 2 km. The uplifted formation on both sides has been eroded. In the basin, secondary anticlines and synclines developed. The Kunlun mountain fault and the Qilian mountain fault thrust to the basin and the maximum offset of the thrusts is over 8 km (Fig. 6).

Discussion. I think this paragraph could be improved in the light of text re-organisation within the previous paragraphs. In particular, I propose to better explain why and how your restoration is different from that already published (e.g., Yin et al., 2008; Zhou et al., 2006).

Answer. Discussion was re-organized, and the difference is: The restoration results of this study is comparable with previous studies by Zhou et al. (2006) and Yin et al. (2008b) in the same area. The total shortening of this study is 25.5 km, which is the shortest, and Zhou et al. gave a 36.61 shortening and Yin et al. got a 41 km result. The difference may be caused by different time-depth conversion and restoration methods, and the total shorting across the Qaidam basin should be about 30 km.

Conclusions (NEW PARAGRAPH). I encourage you to introduce a conclusive paragraph in which you can stress on your main results and implications (at the scale of the Qaidam basin and at the scale of the Tibet plateau).

Answer. Conclusions were added: 3D modeling shows that both sides of the mountains thrust to the Qaidam basin, and the basin is squeezed to a narrow irregular diamond shape. The tectonic deformation in the north, where the Altyn Tagh fault lies, is stronger than in the south. The Altyn Tagh fault, as a boundary strike-slip fault, controls the formation and evolution of the Qaidam basin. The amount of slip southward transformed into thrusts and folds, and the total shortening and shortening ratio during Cenozoic respectively reaches 25.5 km and 11.2% across the basin. The compression of Qaidam basin was not uniform, and it experienced an increase and a decrease, then an increase process. There are two uplift mechanisms of northeastern Tibet, one is the oblique uplift of the whole block along the Altyn Tagh fault, and another is thrusting and

folding that caused thickening of the upper crust in the Qaidam basin and surrounding areas.

Minor Comments

- Page 2, lines 1-4: too long phrase. Please, split it into two concise phrases.

Answer. The phrase has been changed: In this study, we constructed the main geological structures and surfaces in three dimensions through the interpolation of regularly spaced 2D seismic sections, constrained by wells data and surface geology of the Qaidam basin in northeast Tibet. Meanwhile the Cenozoic tectonic history of the Qaidam basin was reconstructed and the uplift mechanism of the Tibetan plateau was discussed.

- Page 2, line 3: add "(northeast Tibet)" after Qaidam basin.

Answer. "in northeast Tibet"was added.

- Page 3, lines 8-10. I don't understand the relations between this statement and the previous one. I suggest to delete it (or remove and to use it within the Discussion).

Answer. It has been deleted.

- Page 3, lines 11-12: Qiliam and Kunlun mountains have been never mentioned before. Not easy to understand their relationships with the Qaidam basin.

Answer. It has been deleted, and Qilian and Kunlun mountain were mentioned in Regional geology.

- Pages 4-5, lines 22-29 and 1-7: all the information should be better supported by a more detailed geological-structural map in Figure 1 (see below) and by field pictures in Figure 2.

Answer. It has been dealt in the Figures.

- Page 5, line 2: delete "to" and change the comma in full stop.

Answer. It has been deleted and the comma has been changed in full stop.

- Page 5, lines 12-19: this is no part of the Results paragraph. It should be moved within Methodology and re-arranged.

Answer. It has been moved and re-arranged in Methods.

- Page 5, line 22: "(Fig. 4)" not appropriate to cite there. Remove.

Answer. It has been removed.

- Page 5, lines 22-25: this phrase is very confusing. I suggest to rewrite it. Anyway, as you are declaring that thrusts are related to the Altyn Tagh fault, I would like to understand if you consider them either (i) fault termination in a horsetail arrangement, or (ii) contractional duplex in restraining bend of the Altyn Tagh fault, or (iii) resulting from restraining stepover between two parallel faults (i.e., the Altyn Tagh fault and the Kunlun fault).

Answer. It has been deleted. I wanted to express that the strike-slip motion transformed into thrusts and folds, whereas it was not clearly described.

- Page 5, line 20 to page 6, line 5: it seems to me that the information derives from previous works. If so, I strongly suggest to delete it from here and to merge it within the paragraph concerning the Regional Tectonic Context.

Answer. It has been deleted and rewritten.

- Page 6, line 8: why do you introduce Fig. 6 before Fig. 5?

Answer. In previous parts, Fig. 5 has been added.

- Page 6, lines 22-25: delete this information and merge it within the Regional Tectonic Context.

Answer. It has been deleted.

- Page 6, line 27: "Means" with lowercase letter.

Answer. It has been changed.

- Page 7, line 3: change "indicating" with "suggesting".

Answer. It has been changed.

- Page 7, lines 5-10: this is part of the Regional Tectonic Context.

Answer. It has been moved to Regional geology.

- Figure 1. I suggest these caprions: ïĆğ (A): Structural map of the Tibet Plateau within the Eurasia/India continental collision. Black arrow indicates present-day displacement of India continental block. ïĆğ (B): Shaded relief map. . . . . .. - Figure 1: provide latitude degrees in the shaded relief map. Provide a legend for explaining the meaning of A-B, C-D, line with triangles (thrust).

Answer. The captions have changed, and legend and locations of other figures have been marked in Figure 1; latitude degrees were added and legends were provided.

- Figure 2: ïĆğ indicate orientation and an approximate scale for all pictures. ïĆğ provide location of these structures in Figure 1. ïĆğ Improve legibility by introducing some labels and symbols indicating bedding (strike and dip), trace of possible fault segments, offsets along fault segments, . . .. ïĆğ Within the caption: change "filed" with "field".

Answer. Orientation and approximate scale have been added; locations have been marked in Figure 1; labels have been improved; "filed"has been corrected.

- Figure 3: ïĆğ Add scale for all figures. ïĆğ In printed page, labels and symbols are not legible. Improve them by changing colour and/or increasing the size. ïĆğ No caption is provided for (A). ïĆğ In (B): which is the meaning of the big red arrow and the black one? Please, indicate the river you are considering. ïĆğ In (E): the stereographic projection is not visible. As suggested above, please provide more accurate structural dataset. ïĆğ Within the caption: change "filed" with "field". ïĆğ The caption of this figure

should be re-organised by following the figure sequence from (A) to (F). ïĆğ Statement in lines 5-6 makes no sense if you don't provide an accurate structural dataset and statistical analysis of the pitch values.

Answer. Approximate scale have been added; labels have been improved; caption has been provided for (a); in (b), the red arrow was deleted, the fault indicated by the dashed red line, and the river was drawn in the image; in (e), the stereographic projection was enlarged, and the result was described in the caption; filed"has been corrected; the caption has been re-organized; last statement was deleted.

- Figure 4: ïĆğ This figure is not explained within the main text. Geometries, depths and offsets of the main seismic reflectors should be described. ïĆğ All red lines are thrusts? Please provide a legend. ïĆğ Indicate the Altyn Tagh fault in seismic profile C-D

Answer. The figure has been explained in the revision; all red lines are trusts, and described in the caption; Altyn Tagh fault was showed in the CD profile.

- Figure 5: ïĆğ Which is the meaning of T0, T21, T4, and TR as reported in caption? Are these time steps? Depths? Please, indicate within the figures. ïĆğ What does the green arrow? North? India displacement?

Answer. They are elevation depth (meter), and green arrow is north direction. The caption was revised.

- Figure 6: following you model, uplift attained only after N23 layer deposition. Indeed, starting from the Abstract, you stated that deformation and uplift were synchronous in Tibet (thus in the Qaidam basin). Please, explain.

Answer. In the restoration, because we did not know exact elevation of the surface except present, we just used 0 meter as a reference, so it shows that the uplift is only after N23. Whereas in the model, the faults and folds moved continuously, and the upper crust was shortened and uplifted at same time.

- Figure 7: what does the green symbol? Does arrow point the north?

Answer. I wanted to show oil wells on the anticline, and in the revision, they have been deleted. The arrow indicates the direction of the compression, and it has been marked in the figure.

I hope that my response can clarify your concern.

Sincerely,

Jianming Guo

Key Laboratory of Petroleum Resources, Gansu Province / Key Laboratory of Petroleum Resources Research Institute of Geology and Geophysics, Chinese Academy of Sciences Lanzhou 730000, China.

gjm2001cn@yahoo.com
* * *
[Figure]

[Figure]

**Fig. 1.** Figure 1: (a): Structural map of the Tibet Plateau within the Eurasia/India continental collision. Black arrow indicates present-day displacement of India continental block. (b): Shaded relief map of

**Fig. 2.** Figure 2: Folds in the Qaidam basin. (a-c) 3D remote sensing images of folds, (d, e) field photos of the limbs of the folds, (f) core of a fold.

[Figure]

[Figure]

**Fig. 3.** Figure 3: Remote sensing images (a, b) and field photos (c, d, e) of the Altyn Tagh fault. (a): Folds in Qaidam basin direct obliquely to the Altyn Tagh fault. (b): A river is left-laterally offset by